# Spatiotemporal observation of surface plasmon polariton mediated ultrafast demagnetization

Yuzhu Fan [1], Gaolong Cao[1], Sheng Jiang[2], Johan Åkerman [3] & Jonas Weissenrieder [1] ✉

Surface plasmons offer a promising avenue in the pursuit of swift and localized manipulation of magnetism for advanced magnetic storage and information processing technology. However, observing and understanding spatio-temporal interactions between surface plasmons and spins remains challenging, hindering optimal optical control of magnetism. Here, we demonstrate the spatiotemporal observation of patterned ultrafast demagnetization dynamics in permalloy mediated by propagating surface plasmon polaritons with sub-picosecond time- and sub-µm spatial- scales by employing Lorentz ultrafast electron microscopy combined with excitation through transient optical gratings. We discover correlated spatial distributions of demagnetization amplitude and surface plasmon polariton intensity, the latter characterized by photo-induced near-field electron microscopy. Furthermore, by comparing the results with patterned ultrafast demagnetization dynamics without surface plasmon polariton interaction, we show that the demagnetization is not only enhanced but also exhibits a spatiotemporal modulation near a spatial discontinuity (plasmonic hot spot). Our findings shed light on the intricate interplay between surface plasmons and spins, offer insights into the optimized control of optical excitation of magnetic materials and push the boundaries of ultrafast manipulation of magnetism.

The interplay between ultrafast light pulses and magnetic order has garnered substantial attention, establishing optical manipulation as the preeminent means for ultrafast manipulation of magnetism[1]. The experimental discovery by Beaurapaire et al. that femtosecond pulsed lasers can induce demagnetization of a Ni thin film on sub-picosecond timescales[2] triggered intense research efforts uncovering numerous mechanisms relevant for laser manipulation of magnetism, e.g. ultrafast demagnetization[3,4], photoinduced angular momentum transfer[5], coherent precession[6], all-optical switching[7,8], and laser-induced magnetic phase transitions[9]. Through manipulation of the amplitude, phase, polarization, and other degrees of freedom of light, magnetic states can now be controlled with unprecedented speed and precision, holding transformative potential for applications in magnetic storage and information processing technology[10].

Surface plasmons (SPs), collective oscillations of surface charge coupled with an electromagnetic field, open up new possibilities for the spatiotemporal control of magnetism with ultrafast light as they allow light to be concentrated to nanoscale volumes at dielectric/metal interfaces[11]. This capability enables selective and localized heating of magnetic materials, central in heat-assisted magnetic recording (HAMR) applications[12,13]. Furthermore, the reciprocal interaction between the subwavelength regime of light and spin has

[1]School of Engineering Sciences, KTH Royal Institute of Technology, Applied Physics, AlbaNova, SE-106 91 Stockholm, Sweden. [2]School of Microelectronics, South China University of Technology, 510641 Guangzhou, China. [3]Department of Physics, University of Gothenburg, Gothenburg, Sweden. ✉e-mail: jonas@kth.se

revealed plasmon-mediated enhancement of both optomagnetic and magneto-optical effects[14–16]. Magnetic plasmonic crystals based on these mechanisms have demonstrated significant potential for applications in ultra-high-sensitivity devices[14], such as magnetic field sensors[17]. Despite numerous studies on the spatiotemporal observation and control of SPs[18], spatiotemporal observation of the interaction between SPs and spins has remained elusive. Such observations are critical for the understanding and optimal control of the localized enhancement of optical excitations in tailored magnetic samples[19].

Using Lorentz ultrafast transmission electron microscopy (LUEM, Fig. 1a), we here demonstrate the spatiotemporal observation of ultrafast demagnetization mediated by propagating surface plasmon polaritons (SPP). Two distinct scenarios for the spatiotemporal distribution of ultrafast demagnetization were compared: one mediated by SPP and the other assuming homogeneous light absorption (in the absence of SPP). In contrast to photon detection techniques based on the magneto-optic effect for the characterization of magnetic materials[20], the magnetic sensitivity in LUEM arises from the local action of the Lorentz force in the Fresnel mode in a non-uniformly in-plane magnetized sample[21]. As a consequence, previous research in LUEM has been limited to observations of the dynamic evolution of intrinsic magnetic textures such as magnetic domain walls, vortices, and skyrmions[22–24]. Here, we combine LUEM with transient optical gratings (TG) to control laser-induced gradients of the in-plane magnetization. This combination allows us to study the spatiotemporal behavior of the ultrafast demagnetization near a plasmonic hot spot of a permalloy ($Ni_{80}Fe_{20}$) thin film sample. The spatial SPP intensity distribution was investigated by photo-induced nearfield electron microscopy (PINEM). The results allow us to formulate a mechanism describing how propagating SPP modulate the spatiotemporal demagnetization distribution.

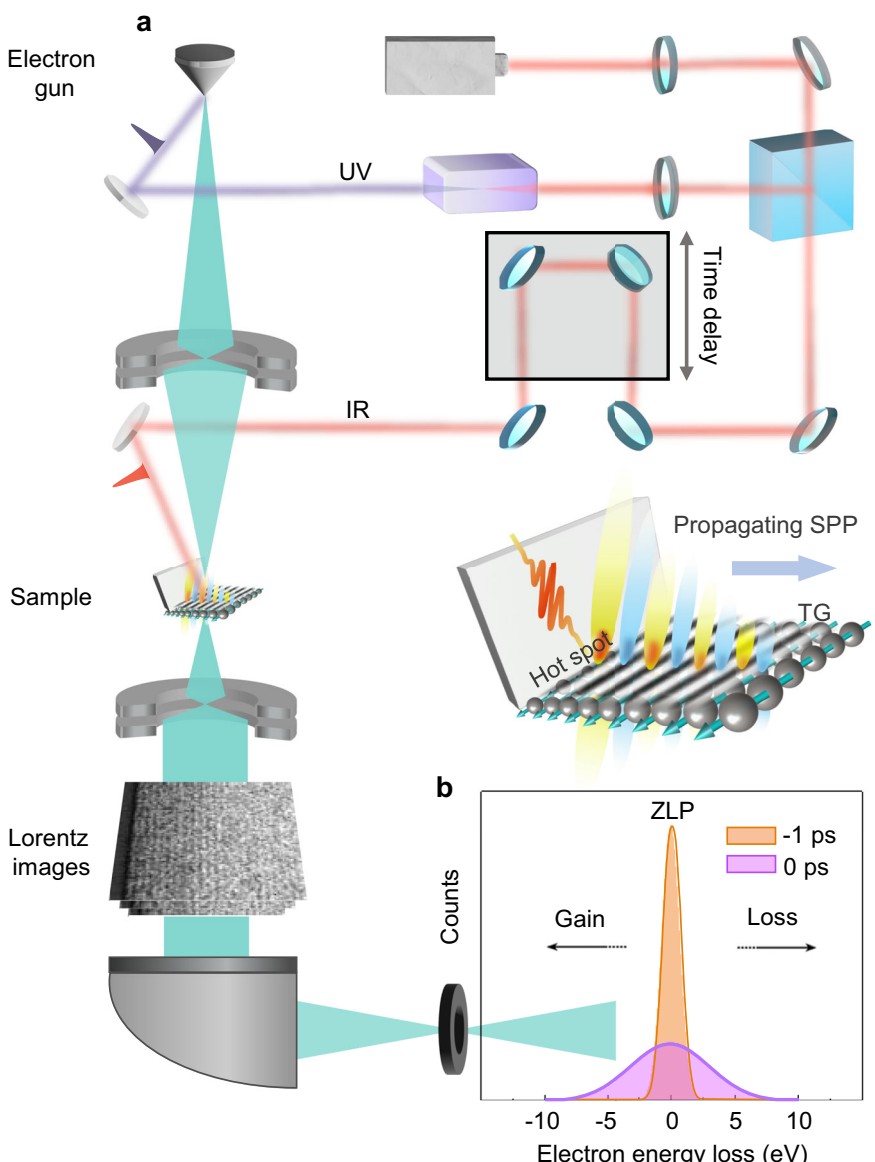

**Fig. 1 | Observation of the interaction between SPPs and spins by LUEM. a** The femtosecond laser is split into two beams, a pump laser beam (red) excites the sample surface, a probe laser beam (purple) hits the cathode, producing photo-emitted electron bunches. Transient optical gratings are generated at the surface of the magnetic sample ($Ni_{80}Fe_{20}$) to control the in-plane magnetization. Hot spot refers to the plasmonic hot spot at the edge of the sample. Transient magnetic gratings (TMG) are observed with LUEM using a retractable camera. The spatial intensity distribution of SPP was determined by analyzing the electron energy distribution after the interaction between electrons and SPP using a post column imaging energy filter mounted under the retractable camera. **b** Fitted PINEM spectra before time zero (electrons have no interaction with SPP) and at time zero (electrons interact with SPP) with IR pump.

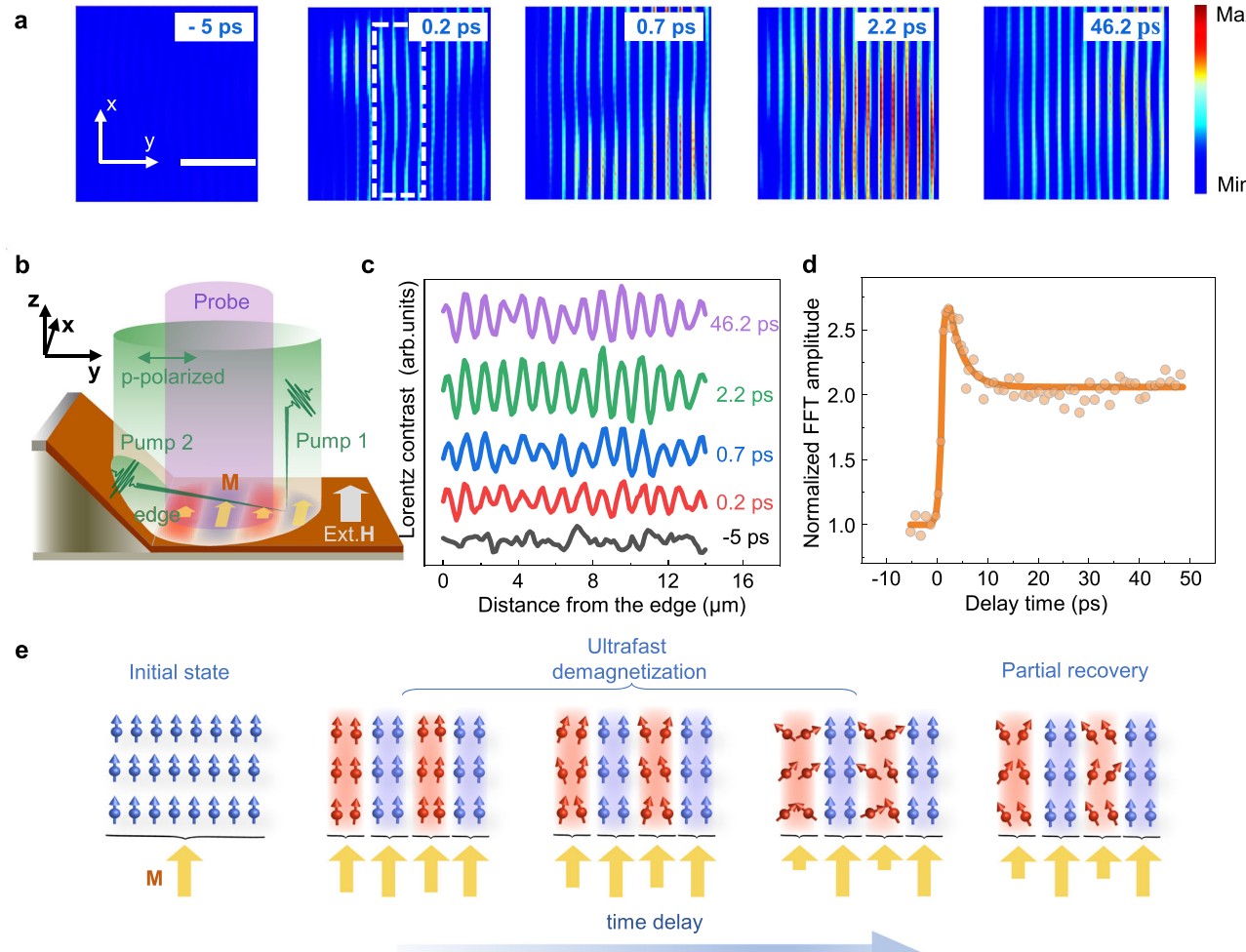

**Fig. 2 | Temporal evolution of the demagnetization process of Ni₈₀Fe₂₀ observed with transient magnetic gratings. a** LUEM images at the temporal delays of −5, 0.2, 0.7, 2.2 and 46.2 ps (FFT filtered at the grating spatial frequency). The scale bar represents 6 μm. **b** Schematic illustration of the experimental configuration to generate TMG in the LUEM geometry. The area enclosed by the green cylinder shows the spatial range covered by the pump beam. Pump 1 indicates the beam directly illuminating the sample and pump 2 represents the fraction of the laser beam reflected from a nearby slanted surface before illuminating the sample. The interference of pump 1 and pump 2 results in a transient optical grating on the sample, where the red gratings represent regions with constructive interference and the light purple gratings represent regions of destructive interference. The

structured excitation will result in local magnetization dynamics that can be observed by LUEM. An external magnetic field is applied with the in-plane component parallel to the sample edge. **c** Line profiles at the same temporal delays as in (**a**) extracted from the raw images with an integration width of 11 μm in a direction perpendicular from the edge of the sample. The traces are offset for clarity. **d** Temporal evolution of the FFT amplitude at the TMG spatial frequency performed in the spatial region indicated by the white dashed box in (**a**). **e** Schematic illustration of the local magnetic response at regions of constructive (red) and destructive (light purple) interference of the optical TG at different stages of the demagnetization process.

## Results

### Observation of ultrafast demagnetization dynamics with transient magnetic grating

Figure 2a shows the dynamic evolution of the transient magnetic grating (TMG) at temporal delays of -5, 0.2, 0.7, 2.2 and 46.2 ps. No magnetic contrast can be observed before time zero, as expected from a uniform alignment of the magnetic moments in a static external magnetic field in excess of the coercive field. The in-plane component of the external field is parallel to the edge, hereafter the edge refers to the position where the membrane and slanted faces of the grid intersect. A magnetic grating contrast appears after time zero and exhibits a maximum at 2.2 ps. Figure 2b shows a schematic illustration of the experimental configuration for the generation of the TMG. The pump laser beam is carefully aligned over the edge in order to illuminate both the slanted face and the membrane. This splits the pump laser beam into two beams, a direct beam (pump 1) and a beam reflected from the slanted face (pump 2). The reflected and direct beams coincide

spatiotemporally and interfere to produce an optical TG on the membrane. The fringes in Fig. 2b illustrate regions of constructive (red) and destructive (light purple) interference. A local demagnetization of the sample will follow almost immediately after photoexcitation, with regions excited by constructive interference fringes experiencing larger demagnetization than deconstructive regions. The relativistic electrons will experience a sinusoidal periodic modulation of the Lorentz force as a result of the TG induced periodic modulation of the local in-plane magnetization[21]. This leads to a modulation of the local deflection angles and results in a sinusoidally periodic TMG contrast after time zero in the LUEM images, as shown in Fig. 2c. The supplementary material (Supplementary Fig. 1) shows a fast Fourier transform (FFT) analysis of the Lorentz contrast at 1.7 ps with a peak at the spatial frequency of 1.0 μm⁻¹, in agreement with the calculated spatial periodicity of the interference fringes of 1.0 μm using an infrared pump wavelength of 1030 nm (1.2 eV). We used an average fluence of 3 mJ cm⁻² in the experiments. For more details of the sample

geometry, the control of the magnetic field direction, and the extraction of the periodicity of the TMG, see method and reference[25].

Figure 2d shows the temporal evolution of the FFT amplitude at the TMG spatial frequency from the Lorentz images in the spatial region indicated by the white dashed box in Fig. 2a. The trace shows how the magnetic grating contrast increases rapidly at time zero, passing a maximum at 2.2 ps, and then gradually decreases. The process from time zero to 2.2 ps corresponds to a photodriven ultrafast local demagnetization. A partial recovery begins already at short time delays. Owing to the space-charge broadening effect on the probe electrons[23], the temporal scale governing the ultrafast demagnetization reflects the constraints imposed by the instrumental time resolution. The mechanism for optically driven ultrafast demagnetization is a topic of intense debate[26,27]. There are several models describing ultrafast dynamic magnetic processes, including the two-temperature model (TTM)[28], the modified two-temperature model with strong electron-spin coupling (s-TTM)[29], the microscopic three-temperature model (M3TM) with angular momentum transfer[26], and the heat-conserving three-temperature model (HC3TM)[30]. However, the three-temperature (3TM) model that includes energy flow between the electron, spin and lattice sub-systems[2] is still effective for a phenomenological explanation of the experimental results. After the initial optical excitation of the electron subsystem, energy swiftly transfers to the spin system via electron-spin coupling, resulting in an ultrafast demagnetization on a femtosecond timescale. Subsequent interactions with the lattice serve as a dissipative pathway, leading to a partial recovery of the magnetization. Fig. 2e provides illustrations of the localized magnetic configurations during the demagnetization and recovery processes. These illustrations schematically depict a sequence of events for which the TMG contrast intensifies and subsequently fades away over time. The evolution is in agreement with the observed changes in the FFT amplitude (Fig. 2d). We use the FFT amplitude to quantitatively characterize the magnitude of the TMG contrast. In order to investigate the relationship between the FFT amplitude and the magnitude of demagnetization, we simulate the spatial laser intensity distribution of the optical TG as a function of the distance from the edge by using a measured 50% reflectivity[25] of the slanted permalloy surface and the optical path difference between the direct pump beam and the reflected pump beam (Supplementary Figs. 2 and 3). The local fluence at neighboring constructive and deconstructive interference fringes differs by an average factor of 14.4 within a 50 µm distance from the sample edge. Consequently, the FFT amplitude, as a function of time delay in Fig. 2d, becomes directly proportional to the absolute demagnetization magnitude at the

regions of constructive interference. The correlation between the FFT amplitude and the demagnetization magnitude allows us to extract the demagnetization time constant $t_{demag}$ and recovery time constant $t_{remag}$, which we will demonstrate later.

## The spatial distribution of Surface Plasmon Polariton intensity measured by Photo-induced Near-field Electron Microscopy

To demonstrate preferential excitation of SPP from the edge and measure the spatial distribution of SPP intensity, we employ photo-induced near-field electron microscopy (PINEM)[31–37], to analyze the electric field-electron coupling strength across the sample surface. The PINEM technique has been used to detect evanescent electromagnetic fields in the vicinity of nanostructures or at a surface, where inelastic electron-photon interaction can take place due to the conservation of momentum[32] and electrons can absorb or lose energy from one or several scattered photons. When the electron probe and the evanescent near-fields spatiotemporally overlap, discrete energy peaks separated by the quantized photon energy of the pump laser will appear as sidebands to the zero-loss peak (ZLP) in the electron energy loss spectrum. Fig. 3a shows a typical PINEM electron energy loss spectrum obtained under the same configuration as the demagnetization experiments (identical relative positions of the sample edge, pump centre position, and sample tilt angle). The front view and side view of the PINEM spectrum reflect the energy of the excited photons (2.4 eV) and the duration of the electron pulses (~0.9 ps), respectively. Detailed technical information is provided in the Methods section. Fig. 3c shows the spatial dependence of the PINEM intensity analyzed at increasing distance from the edge, indicated by the red dashed circles in Fig. 3b. The PINEM intensity was calculated from an integration of the $\hbar\omega$ peak at time zero, which is approximately proportional to the square modulus of the surface plasmonic near-field[32,38,39]. As illustrated in Fig. 3c, the spatial dependence of the PINEM intensity follows an exponential decay with the distance from the edge and an exponential fit yields a characteristic decay length of approximately $15 \pm 3$ µm. From the analysis of the surface plasmonic near-field, we observe that the PINEM intensity close to the sample edge is approximately twice of what is found at a 40 µm distance from the edge. As reported in previous PINEM studies[38], the near-field intensity of the surface plasmon will experience a linear dependence with absorbed laser fluence in the range of local fluences used in our experiments, indicating that the spatial distribution of the PINEM signal in our case characterizes the local absorption of the pump laser.

The SPPs give rise to evanescent fields, which decay spatially in the direction perpendicular to the sample surface. A SPP confined at

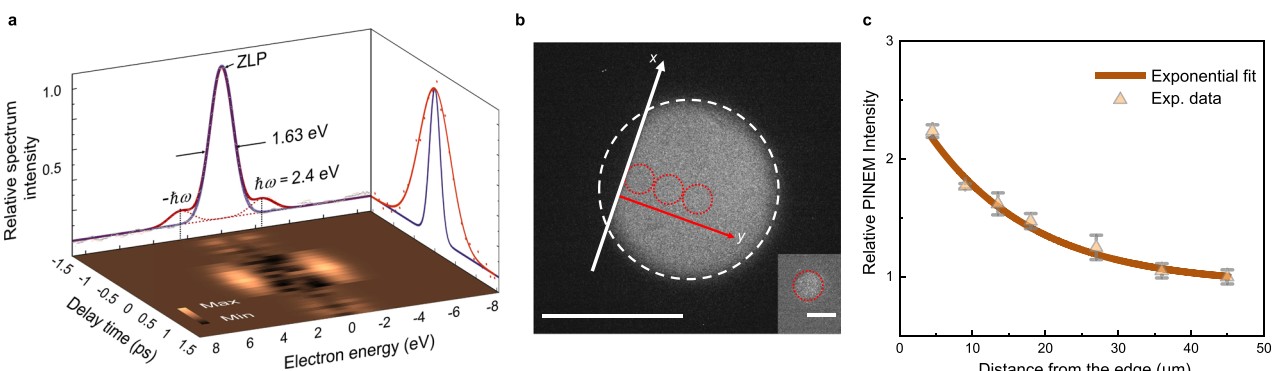

**Fig. 3 | Mapping the spatial distribution of the PINEM intensity. a** The front view shows a typical PINEM electron energy loss spectrum at time zero. The top view shows the temporal evolution of the electron energy loss spectrum (zero loss peak subtracted). The side view shows the temporal trace of the integration of the $\hbar\omega$ peak intensity fitted by a Gaussian function (0.88 ps, red line) and the temporal profile of the laser pulse (0.3 ps, blue line). **b** Sample geometry with the edge indicated by a white solid line. The PINEM intensity is analyzed within the red dashed circles along the direction marked by the red arrow. The scale bar indicates 50 µm. The inset shows the image of the focused electron beam. The scale bar in the inset indicates 10 µm. **c** Detected relative PINEM intensity as a function of distance from the edge of the sample, from integration of the $\hbar\omega$ peak at time zero. The error bars represent the standard deviation of ten repetitive experiments.

the sample surface of a material with the magnetic permeability $\mu \neq 1$ propagates with the following dispersion relation[11]:

$$\mathbf{k}_{spp}(\omega) = \frac{\omega}{c} \sqrt{\frac{\varepsilon_1(\omega)\varepsilon_2(\omega)}{\varepsilon_1(\omega)+\varepsilon_2(\omega)}}(1+\alpha g_x) \qquad (1)$$

Where $\mathbf{k}_{spp}$ is the SPP wave vector, $\omega$ is the incident light frequency, $c$ is the speed of light in vacuum, $\varepsilon_1$ and $\varepsilon_2$ are the dielectric constants of vacuum and permalloy, respectively, $\alpha = (-\varepsilon_1\varepsilon_2)^{-1/2}(1-\varepsilon_2^2/\varepsilon_1^2)^{-1}$, $g_x$ is the $x$-component of the gyration vector (a magneto-optical parameter) $\propto$ magnetization[11]. Supplementary Fig. 4 shows results from calculations of the dispersion relationships of the two modes of SPP excited at the top (Air/Al (5 nm)/$Ni_{80}Fe_{20}$ (50 nm)) and bottom ($Ni_{80}Fe_{20}$ (50 nm)/$Si_3N_4$(50 nm)/Air) interfaces of the sample by solving Maxwell's equations in the frequency domain, using the Radio Frequency Module in the COMSOL Multiphysics® software based on the implemented Finite Element Method (FEM). The contribution of SPP from the bottom interface can be neglected due to the strong light absorption in the sample (Supplementary Fig. 4e). The spatial discontinuity at the edge can facilitate momentum conservation and enhance light-sample coupling[38–42], serving as a plasmonic hotspot. This result in the excitation of SPP at the edge and their propagation perpendicular from the edge, leading to an exponential spatial distribution of SPP intensity along the propagation direction[39,40].

## Mapping the spatiotemporal demagnetization dynamics with SPP mediation

Since the SPP is preferentially excited at the plasmonic hotspot (edge) we can investigate the SPP mediated contribution to the demagnetization dynamics. We proceed by dividing the Fresnel images obtained at a specific time delay into a series of slices parallel to the edge (Fig. 4a) to analyze the ultrafast demagnetization dynamics as the function of distance from the edge. By performing a spatial FFT for each slice and time delay, we can extract the FFT amplitude associated with the spatial frequency of the TMG. This allows us to generate time traces of the demagnetization dynamics at increasing distance from the edge. Fig. 4b presents representative demagnetization curves corresponding to spatial locations of the slices illustrated in Fig. 4a. In the current analysis, our focus is on the ultrafast demagnetization processes occurring within a few picoseconds after time zero.

After fitting the demagnetization curves by a bi-exponential function, $t_{demag}$, $t_{remag}$, as well as other fitting parameters can be extracted as a function of distance of the slice centre to the edge, as shown in Supplementary Fig. 5. The obtained demagnetization time constant $t_{demag} = 0.117 \pm 0.015$ ps is similar to reported demagnetization time constants for this material from photon-probe techniques such as X-ray Magnetic Circular Dichroism (XMCD)[43] and Magneto-optical Kerr effect (MOKE)[44,45]. Figure 4c shows the experimental space-time contour plot of the FFT amplitude, at the spatial frequency of the TMG, and at increasing distance from the edge. From this space-time contour, we observe distinct spatiotemporal characteristics of the demagnetization process. These include an increased demagnetization magnitude near the edge, an additional magnetic periodicity beyond what is expected from the TG, and a delay in reaching maximum demagnetization magnitude with increasing distance from the edge. Notably, the spatiotemporal distribution of demagnetization examined in Fig. 4c is influenced by both the sinusoidal light intensity distribution from the optical TG and the intensity distribution from SPP excited at the edge. To separate these effects, we simulated the influence of the sinusoidal spatial light distribution on the spatiotemporal demagnetization distribution and plotted the space-time contour of demagnetization controlled solely by the TG, as shown in Supplementary Fig. 6f. The simulation is performed by a method involving atomistic spin dynamics (ASD) simulations implemented in the Vampire software package[46]. We simulate the local

demagnetization dynamics from the optical TG intensity distribution (Supplementary Fig. 3), with technical detailed presented in Supplementary Fig. 6. In Fig. 4d, the spatial distributions of the relative maximum demagnetization amplitude $\Delta F_{max}$ (the difference between the maximum FFT amplitude and the average FFT amplitude before time zero, see Fig. 4b) were extracted from the space-time contours of both Fig. 4c and the reference simulation (Supplementary Fig. 6f) and fitted with exponential functions.

Comparing the spatial distribution of relative $\Delta F_{max}$ under the influence of the SPP with the simulated reference reveals that $\Delta F_{max}$ is almost twice as large near the edge compared to at a 50 μm distance (orange circle data points in Fig. 4d) in the SPP-mediated case. In contrast, in the simulated reference data, $\Delta F_{max}$ increases only by a factor of 1.4 near the edge compared to at 50 μm distance (green triangle data points in Fig. 4d). This shows that the presence of SPP enhances the demagnetization. More strikingly, the presence of SPP introduces an additional magnetic period different from that of the TG. We observe enhancements of $\Delta F_{max}$ at spatial intervals of approximately 17 μm (orange circle data points in Fig. 4d). This effect resembles a beat and can be observed at data points deviating from the exponential fit in Fig. 4d, most evident at around 13 μm and 30 μm. The periodicity of the magnetic beat is absent in the simulated reference data, indicating its direct association with SPP. To determine the periodicity of the magnetic beat, we extracted the average FFT amplitude within the 1-3 ps time delay range as a function of the distance from the edge (Fig. 4e). After subtracting a background fitted with a single exponential function, an FFT of the trace reveals a spatial frequency for the magnetic beat at around 0.06 μm⁻¹ (inset of Fig. 4e), or a period $\lambda_{beat}$ of ∼17 μm. To isolate the contribution of SPP to the spatio-temporal distribution of demagnetization, we conducted control experiments using a sample geometry excluding the plasmonic hotspot and eliminating SPP excitation. In the absence of the plasmonic hotspot, the optical absorption of the sample is spatially homogeneous, and the demagnetization dynamics are solely governed by the intensity of the optical TG. A schematic illustration of the control experiment's geometry is shown in Supplementary Fig. 7. A comparison of the demagnetization results, with and without the influence of SPP excitation, is shown in Fig. 4f. To clarify the SPP contribution to demagnetization, we subtracted the simulated reference data from the analysis of both demagnetization results. This subtraction eliminates the influence of the TG distribution on the demagnetization. In the absence of SPP-mediated ultrafast demagnetization, we found that the spatial distribution of relative $\Delta F_{max}$, after subtraction of the simulated reference, is close to zero (blue curve in Fig. 4f). This indicates that the demagnetization dynamics are governed solely by the interference of the pump light beams, and it also serves to validate the simulated reference results. However, in the presence of SPP-mediated ultrafast demagnetization, the spatial distribution of relative $\Delta F_{max}$, after subtraction of the simulated reference, is significantly enhanced near the edge. This clearly demonstrates the contribution of excited SPP to the local demagnetization (orange data in Fig. 4f). Additionally, the ∼17 μm magnetic beat signature is only present in the experiments involving SPP excitation. Taken together, these results confirm the enhanced demagnetization and modulation of the spatial distribution of demagnetization induced by the excitation of SPP.

In addition to above findings regarding the spatiotemporal distribution of the demagnetization with contribution from SPP modulation, we also observe a spatial delay of the time required for reaching maximum demagnetization. The time corresponding to highest FFT amplitude at each spatial position in Fig. 4c exhibits a tilt, also observed in Fig. 4b, indicating that the time to reach maximum demagnetization increases with distance from the edge. Supplementary Fig. 8 provides further evidence that the time required to reach maximum demagnetization is increasingly delayed by up to 0.3 ps over a distance of 50 μm from the edge, as directly observed from the

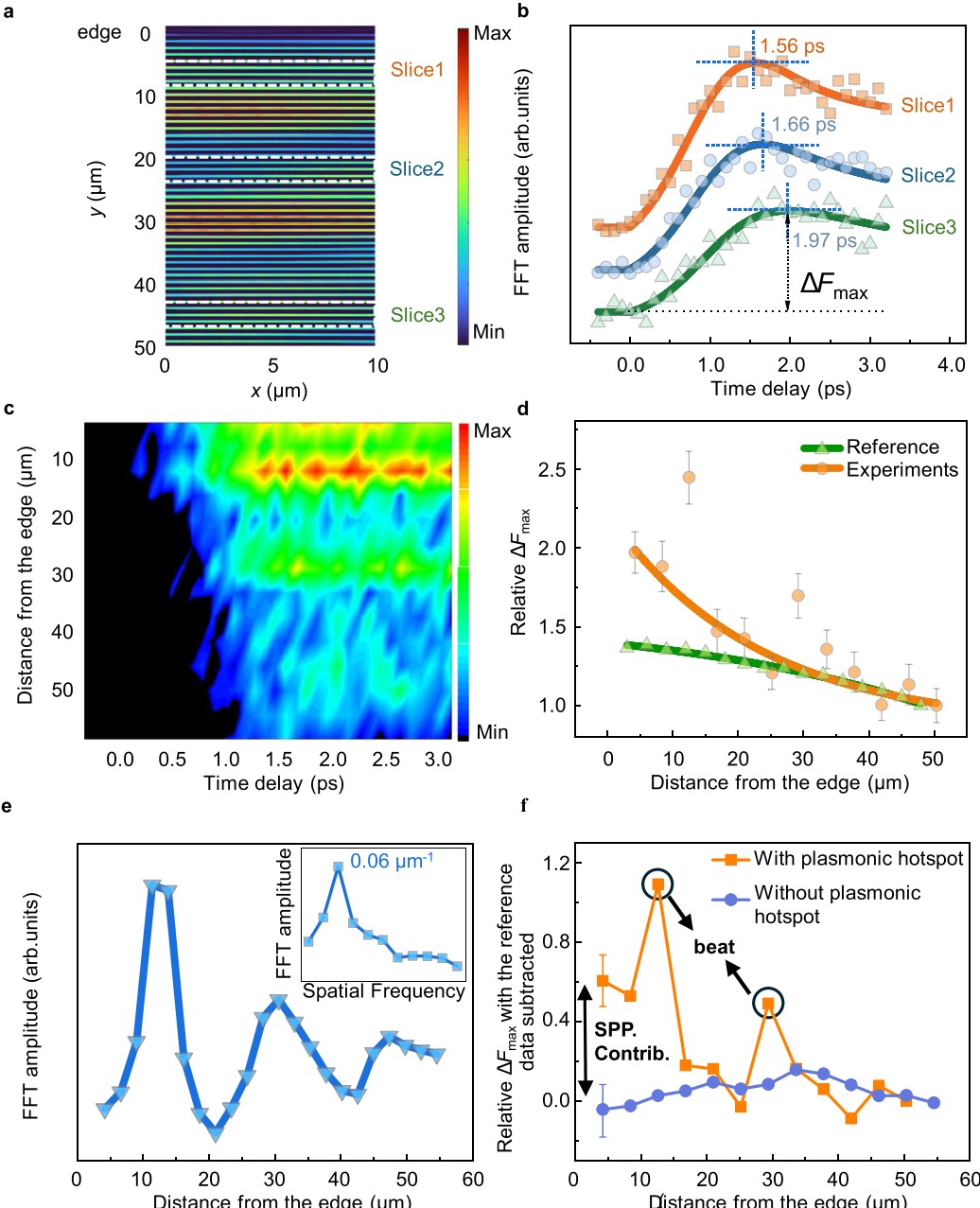

**Fig. 4 | Spatiotemporal mapping of demagnetization dynamics near the edge.**
**a** A typical LUEM image at a time delay of 2.9 ps. The image is FFT filtered at the spatial frequency of the TMG. **b** Selected demagnetization time traces from slices with the same color code as in (**a**). The traces are offset for clarity. The FFT amplitude is obtained at the spatial frequency corresponding to the TMG. The data points are the experimental results, and the solid lines are the result from fitting. **c** Experimental space-time contour of the FFT amplitude as a function of distance from the edge. **d** Experimental (orange) and simulated reference (green) relative $\Delta F_{max}$ as a function of distance from the edge, where $\Delta F_{max}$ is maximum demagnetization amplitude, characterized by the difference between the maximum FFT amplitude minus the average FFT amplitude before time zero as shown in (**b**). Solid

lines are fitted by exponential functions. The reference data refers to simulated distribution of relative $\Delta F_{max}$ dominated by TG. **e** Averaged FFT amplitude over 1-3 ps time delay as a function of distance from the edge. The inset shows the spatial frequency of the beat in (**e**) after FFT processing. **f** The spatial distribution of the $\Delta F_{max}$ with plasmonic hotspot (orange data) and without plasmonic hotspot (purple data). To isolate the contribution of SPP to the demagnetization, we subtracted the simulated reference data for the analysis of both demagnetization results. The vertical arrow refers to the SPP contribution. The error bars in this figure represent the standard deviation of the measured demagnetization data from the bi-exponential fit.

original LUEM images after filtering. Control experiment, with no SPP hotspot, also show that the maximum demagnetization time near the edge is faster (approximately 0.2 ps) than at a distance of ~ 50 μm from the edge (Supplementary Fig. 9). We calculated the effect of the optical path difference between the reflected and incident beams on the relative time delay of the interfered pulse center (in time) and pulse broadening, both of which influence the spatial distribution of the

maximum demagnetization time. For a detailed discussion, please refer to Supplementary Fig. 10. Our calculations show that a 50 μm optical path difference results in a ~ 0.2 ps delay in the maximum demagnetization time, which aligns with the spatial delay observed in the control experiment in Supplementary Fig. 9. The results from the control experiment, without a plasmonic hotspot, indicate that the fitted slope of the spatial delay curve for the maximum

demagnetization time shows no significant difference compared to the experimental results including a plasmonic hotspot, as shown in Supplementary Fig. 9. This suggests that the spatial delay in the maximum demagnetization time observed in Fig. 4b, c and Supplementary Fig. 8 is primarily due to the optical path difference from the optical TG.

## Discussion

We observe a correlation between the spatial distribution of the SPP intensity (Fig. 3c) and the maximum demagnetization magnitude $\Delta F_{max}$ (Fig. 4d), when analyzed under the same sample geometry. This correlation is illustrated in Supplementary Fig. 11a where the curves are plotted in the same graph and further confirmed by the simulated spatial distribution of SPP intensity exhibited in Supplementary Fig. 4f, which is consistent with the measured spatial distribution of SPP intensity shown in Fig. 3c. Next, we use an effective pump fluence scaled to the measured spatial PINEM intensity distribution in Fig. 3c as input to ASD simulations of the spatial distribution of $\Delta F_{max}$ using the Vampire software package[46] (Supplementary Fig. 11b–d). The resulting simulated spatial distribution of is now approximately two times larger near the edge compared to the inner parts of the sample and consistent with the experimental spatial distribution of $\Delta F_{max}$ in the presence of SPP in Fig. 4d, as shown in Supplementary Fig. 11d. Moreover, the simulated characteristic lifetime of the spatial distribution of maximum demagnetization magnitude $\Delta F_{max}$ is approximately $20 \pm 4$ μm from the exponential fit, similar to the mean lifetime of approximately $23 \pm 10$ μm from the exponential fit of the experimental results, as shown in Supplementary Fig. 11d. Our analysis shows that the spatial distribution of SPP intensity determines that of the maximum demagnetization magnitude near a plasmonic hotspot. This is expected as the maximum demagnetization amplitude is proportional to the square modulus of the local absorbed electric field intensity[47], including the SPP intensity, and demonstrates the mechanism by which SPP enhances demagnetization and modulates the spatial distribution of demagnetization (as shown in Fig. 4f) through changes in the local light absorption.

We observe a magnetic periodicity in the spatial distribution of the demagnetization (Fig. 4e) that is significantly larger than the optical transient grating (TG) period. This larger magnetic periodicity resembles a beat pattern, indicating that the local demagnetization is modulated by two propagating waves: the optical pump wave and the in-plane propagating SPP. Supplementary Fig. 12a show a schematic illustration of the geometry, where the SPP propagates in a direction parallel to the in-plane component of the reflected pump beam, resulting in a co-propagation mode[48–50]. As a result, the electric field of the reflected pump and the SPP at the top interface are superimposed along the propagation direction. The phase velocity mismatch will result in a modulation of the total field with a characteristic beat period[48] $\lambda_{beat} = 2\pi/\Delta k_y$, where $\Delta k_y$ is the difference in momentum between the in-plane component of the reflected light ($k_{ypump}$) and the SPP wave packet ($k_{ySPP}$). Therefore, the superposition of the SPP and the laser at the top interface will result in an optical beat. By extracting the wave vectors of SPP and free light from the simulated dispersion relationships, the periodicity of the optical beat is calculated as 16.81 μm, essentially identical to the experimentally observed magnetic periodicity of ~17 μm. Details of the simulations are provided in Supplementary Fig. 12b. We can, therefore, conclude that the observation of a beat associated with an enhanced demagnetization at a periodicity $\lambda_{beat}$ is due to the superposition of co-propagating optical and SPP waves at the top interface, further demonstrating the interaction between propagating SPP and demagnetization dynamics.

In conclusion, we have performed spatiotemporally analysis of optically structured (TG) ultrafast demagnetization dynamics near a plasmonic hotspot. Through experiments and simulations, we isolated the contributions of direct TG excitation and SPP to the spatiotemporal

distribution of ultrafast demagnetization. Our findings demonstrate that propagating SPP modulates the spatiotemporal distribution of ultrafast demagnetization process in terms of demagnetization amplitude and period. Several reports describe how plasmonic nanostructures, including nanoantennas and metasurfaces on magnetoplasmonic crystals, can enhance the magneto-optical response, modify demagnetization dynamics, and improve demagnetization efficiency by tuning the SP resonance[11,19,51–55]. By introducing transient optical grating and tailoring the sample geometry, we can locally modulate the light intensity distribution, enhance excitation, and ultimately control the ultrafast magnetic response on sub-picosecond timescales and sub-μm spatial scales, even off the SP resonance. We envision that the results from our work, combining high spatial resolution and structured light to determine magnetization dynamics, will encourage further exploration of ultrafast light-magnetization interactions at nanoscale spatial dimensions. This includes areas such as all-optical switching, THz nanomagnonics, and interactions between structured light and magnetic topological textures.

## Methods
### Sample geometry and preparation
The permalloy ($Ni_{80}Fe_{20}$) thin film sample with a thickness of 50 nm was deposited on a $Si_3N_4$ supporting membrane with a thickness of 50 nm (Ted Pella, USA). The membrane window size was 100 μm × 1500 μm, with the long edge of the window being carefully positioned parallel to the x axis as shown in Fig. 2b. The slanted faces were angled at 35° relative to the membrane normal. The incidence angle of the reflected beam to the sample surface was 73°. The supporting membrane was electron transparent and for dedicated use as transmission electron microscopy support grid. The $Si_3N_4$ membrane grid was positioned with the slanted side faces facing towards the target during the deposition process. The permalloy thin film was deposited with magnetron sputtering using an argon working pressure of 2.5 mTorr. The permalloy sample was capped with a 5-nm Al to prevent oxidation.

### Lorentz mode ultrafast electron microscopy (LUEM)
The LUEM used in this work is based on a modified JEOL JEM 2100 TEM integrated with a fiber-amplified laser with a 1030 nm fundamental wavelength and <300 fs pulse duration (Tangerine, Amplitude Systèmes). The sample was excited by 300 fs laser pulses ($\lambda = 1030$ nm, $f = 20$ kHz at an average laser fluence of around 3 mJ cm$^{-2}$. The membrane and the slanted side faces of the grid are positioned in the LUEM facing the direction of incidence of the electron and pump laser beams. A LaB$_6$ guard ring cathode was excited with fourth harmonic generation UV laser pulses ($\lambda = 258$ nm) from the same laser source. To prevent the magnetization of the soft permalloy sample from becoming saturated, the LUEM was operated with the objective lens deactivated. The external magnetic field was tuned by applying a current to the objective lens, and its magnitude for relevant objective lens currents was calibrated using a Hall probe TEM holder. The applied magnetic field direction is perpendicular to the sample surface at zero sample tilt angle, with the option of introducing an in-plane component parallel to the edge of the sample through tilting the sample[25].

The time delay between the pump and probe pulses was controlled by a motorized delay stage, enabling adjustments to the optical path length for the pump pulses. Ultrafast Lorentz micrographs were captured using a CheeTah-Event CF100 camera, based on a Timepix3 chip from Amsterdam Scientific Instruments, the Netherlands. The electron pulse duration used for the LUEM analysis in Fig. 3 was 1.2 ps, as obtained from PINEM analysis using the same experimental conditions.

### Photon-Induced Near-field Electron Microscope (PINEM)
The PINEM in this work was conducted on the same TEM, laser optics and delay stage as for LUEM analysis. The sample was excited by 300 fs green light pulses ($\lambda = 515$ nm, $f = 80$ kHz at the average laser fluence of around 3 mJ cm$^{-2}$. The zero-loss peak exhibited a FWHM of 1.6 eV in the

experiment. The electron energy loss spectroscopy was conducted using a post column imaging energy filter (Gatan Quantum SE).

The front view in Fig. 3a shows the PINEM spectrum at time zero. The red trace in the side view of Fig. 3a shows a temporal trace of the integrated intensity of the $\hbar\omega$ peak as fitted by a Gaussian function. The FWHM, providing an estimate for the electron pulse duration[38] during the PINEM experiments, is estimated to be 0.9 ps. The laser pulse used for excitation, with a width of around 0.3 ps, is shown in blue. All the datasets are recorded with a constant exposure time. To rule out any influence from fluctuations in the electron flux, each dataset is normalized to the integral value of the entire PINEM spectra at time zero, and then averaged over ten acquisitions.

### Atomistic spin simulations of ultrafast demagnetization dynamics

To simulate the ultrafast demagnetization dynamics of permalloy as a function of laser intensity, we conducted atomistic simulations of the response of $10^4$ ferromagnetically exchange-coupled $Ni_{80}Fe_{20}$ spins to an ultrashort 300 fs laser pulse. The demagnetization dynamics was obtained by solving the stochastic Landau–Lifshitz–Gilbert (LLG) equation through the Heun method as implemented in the Vampire software. The atomistic model and input parameters of $Ni_{80}Fe_{20}$ can be found in reference[43,46]. The effect of laser heating on the electronic temperature is simulated using the two-temperature model[56].

## Data availability

All data needed to evaluate the conclusions in the paper are provided in the paper and/or the Supplementary Information.

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

## Acknowledgements

This research was funded by the Knut and Alice Wallenberg Foundation (2012.0321 and 2018.0104), the Swedish Research Council (VR), and through the ARTEMI national infrastructure (VR 2021-00171 and Strategic Research Council (SSF) RIF21-0026) (JW).

## Author contributions

Y.F. and G.C. conducted experiments; J.S. and J. Å. fabricated samples, Y.F. performed simulations; G.C. and J.W. conceived and designed the research; Y.F. and J.W. wrote the article. All authors have read and approved the final manuscript.

## Funding

## Competing interests

The authors declare no competing interests.
