## [Transparent Peer Review file · Nature Communications]

Spatiotemporal observation of surface plasmon polariton mediated ultrafast demagnetization

Corresponding Author: Professor Jonas Weissenrieder

Version 0:

Reviewer comments:

Reviewer #1

(Remarks to the Author)

Fan et al. reported the spatiotemporal imaging of ultrafast demagnetization dynamics in permalloy Ni80Fe20 induced by surface plasmon polariton near field using Lorentz ultrafast transmission electron microscopy and photo-induced near field electron microscopy. Through the transient grating excitation, they recognize a particular spin dynamics that are periodically modulated in space, which nicely corresponds to the period of the transient grating. Based on the analysis of the spatiotemporal Fourier amplitude and numerical modelling, they further assign such modulated spin dynamics to surface plasmon excitation because of the stronger Fourier amplitude and the delayed rise in the dynamics, which cannot be reproduced from simulations using optical excitation only. They finally confirm the plasmonic excitation by mapping the near field plasmonic field intensity, as well as finite difference based numerical simulation. I think the observations reported in this manuscript may be of interest, but results and analysis do not support well the conclusions. Therefore, I cannot recommend it for publication.

Major comments:

1. The authors have been trying to claim the spin dynamics is triggered by plasmonic field. The major argument is that the experimentally observed Fourier component is stronger compared to that observed in numerical simulation based on optical excitation. While there is a small difference, it is not a conclusive evidence of plasmonic enhancement to the spin excitation. Not to mention that the plasmonic propagation time is much shorter compared to the observed delay in the spin rising dynamics, as also pointed out by the authors.

In fact, one can simply “quench” the SPP field by switching the polarization of the pump light to be surface parallel, in which case only the optical grating exists and no SPP should be excited. In this case, one is allowed to compare the TM and TE excitation, and differentiate the plasmonic contribution in space and time. I think this is a more direct evidence of plasmonic effect than the numerical simulation and analysis in the rising time.

2. The conclusion of plasmon induced spin currents is also far reaching. One cannot simply rule out the propagation effect and assign the delayed rising to be due to the spin currents. The spin of SPPs is locked with their momentum, so if an opposite spin configuration is excited by counter propagating SPPs, then one can propose the spin current mechanism. I suppose this could be done with a Lorentz TEM. Simply arguing from the intensity and time perspective is rather weak.

Minor comments:

The authors should treat abbreviations more carefully. LUEM, UEM, LUTEM are used interchangeably. In addition, some concepts are not well defined, and some are repeatedly defined in the main text. This hinders efficient reading of the manuscript.

Reviewer #2

(Remarks to the Author)

Version 1:

Reviewer comments:

Reviewer #1

(Remarks to the Author)

I appreciate that Fan et al. have done further experiments to support their conclusions, and have significantly revised their manuscripts.

Regarding queching of SPPs, although the experiment was conducted in a different way than I suggested, i.e. by varying the excitation polarization, it still serves the same purpose in practice. So it is indeed interesting to see the effect of the enhancement of the spin dynamics with and without SPP excitations, and particularly the beating effect.

As to the delayed spin response, the authors changed their argument to broadening effect of the SPP pulse. This argument seems reasonable qualitatively, but I think it is a material specific question (also the authors compared the results with that in Ag), and needs to be considered along with the propagation effect in my first comment.

With this said, I think it is better to calculate the wavelength, the propagation length, as well as the chirp of the SPPs, based on the dielectric function of the magnetic film. This calculated propagation length will further strengthen the argument for Fig. 3c, and the chirp of SPPs will support whether 100 fs broadening is the correct value to consider.

Lastly, if SPP broadening was considered, perhaps the broadening of the excitation needs to be considered as well.

If the authors could supplement these results to further the current manuscript, I think it will be appropriate for publication.

Version 2:

Reviewer comments:

Reviewer #1

(Remarks to the Author)

The authors have supplemented their experimental observations with further calculations. While the conclusion, which is kind of expected, is somewhat different than before, their experimental observation on the spin beating is still interesting to the community. With the interpretation being corrected and updated, I therefore think it is appropriate for publication.

Reviewer #1 (Remarks to the Author):

Fan et al. reported the spatiotemporal imaging of ultrafast demagnetization dynamics in permalloy Ni₈₀Fe₂₀ induced by surface plasmon polariton near field using Lorentz ultrafast transmission electron microscopy and photo-induced near field electron microscopy. Through the transient grating excitation, they recognize a particular spin dynamics that are periodically modulated in space, which nicely corresponds to the period of the transient grating. Based on the analysis of the spatiotemporal Fourier amplitude and numerical modelling, they further assign such modulated spin dynamics to surface plasmon excitation because of the stronger Fourier amplitude and the delayed rise in the dynamics, which cannot be reproduced from simulations using optical excitation only. They finally confirm the plasmonic excitation by mapping the near field plasmonic field intensity, as well as finite difference based numerical simulation. I think the observations reported in this manuscript may be of interest, but results and analysis do not support well the conclusions. Therefore, I cannot recommend it for publication.

We thank the reviewer for valuable comments and suggestions to our manuscript. We appreciate the time and effort you have dedicated to reviewing our work.

In our revised, and in our view much improved, version of the manuscript we have included additional experimental results that show the demagnetization dynamics without a plasmonic hotspot, i.e. quenched excitation of surface plasmon polariton (SPP). This new result serves as a control experiment that unequivocally demonstrates the influence of SPP on the spatial demagnetization dynamics. We have also changed the outline of the manuscript for an improved reading experience.

In brief our study includes experiments (coupled with simulations) that explore the influence of propagating SPPs on the ultrafast demagnetization dynamics. We demonstrate the existence of SPP excitation by mapping of the near-field plasmonic intensity (through PINEM). In our experiments we find: (1) The spatial distribution of SPP intensity and maximum demagnetization amplitude exhibit a high degree of correlation, (2) the emergence of an additional magnetic periodicity different from the periodicity of transient optical grating (TG). Both these findings provide strong evidence that SPP plays a key role in modulating the demagnetization process. In the revised version we now also present comparative experimental results showing the demagnetization distribution with and without excited SPP. These new results provide direct experimental evidence for the influence of SPP on the demagnetization, rather than relying on comparisons between experiments and simulations as in our original manuscript.

We believe that the revised manuscript, with the additional experimental results and analysis, provides a stronger substantiation of the conclusion made of SPP-mediated demagnetization dynamics.

Major comments:

1. The authors have been trying to claim the spin dynamics is triggered by plasmonic field. The major argument is that the experimentally observed Fourier component is stronger compared to that observed in numerical simulation based on optical excitation. While there is a small difference, it is not a conclusive evidence of plasmonic enhancement to the spin excitation. Not to mention that the plasmonic propagation time is much shorter compared to the observed delay in the spin rising dynamics, as also pointed out by the authors.

We agree with your point that the Fourier amplitude of demagnetization alone does not serve as conclusive evidence for the modulation of demagnetization dynamics by plasmonic excitation. However, in our original manuscript we extended our analysis beyond the Fourier amplitude of

demagnetization. For improved clarity, we have reorganized the results and analysis sections to better convey the message that the demagnetization dynamics are indeed influenced by the plasmonic field, as shown below:

- (1) The correlation between the spatial distribution of SPP intensity and maximum demagnetization amplitude ΔF_{max} , as shown in Fig. R1a. We further employed the concept 'effective pump fluence' where we scaled the absorbed fluence to the measured SPP intensity distribution as input for simulations of the spatial distribution of ΔF_{max} . The simulated spatial distribution of ΔF_{max} is in excellent agreement with the measured spatial distribution of ΔF_{max} when including SPP, as shown in Fig. R1b. Our analysis indicates that the spatial distribution of the SPP intensity determines the maximum demagnetization magnitude ΔF_{max} near a plasmonic hotspot, which serves as evidence that SPP induced demagnetization must be considered to accurately explain the demagnetization dynamics.

Figure R1 (a) The measured SPP intensity distribution from photo-induced near-field microscopy (wine data) and the measured maximum demagnetization magnitude ΔF_{max} distribution (orange data). The graphs are offset for clarity. (b) Simulated ΔF_{max} distribution using the effective pump fluence scaled from the SPP intensity distribution in (a), consistent with the measured ΔF_{max} distribution near the plasmonic hotspot. Since PINEM is only sensitive to SPP intensity, two datapoints in ΔF_{max} distribution (at 13 and 30 μm) corresponding to the beat has been removed here for improved visualization.

- (2) We observed an additional magnetic periodicity of 17 μm , which we refer to as a beat in the original manuscript. This periodicity is distinct from the magnetic grating induced by the transient optical grating that has a periodicity of 1 μm . The presence of this additional periodicity in the magnetic grating substantiates that the excitation of spin dynamics requires not only the wave of the pump light but also another wave to produce the observed superimposed periodicity. In the discussion section of the original manuscript, we explore the origins of this additional magnetic periodicity, noting that its periodicity can be perfectly explained by the expected beat frequency/periodicity of the co-propagating SPP and pump light beam.

Taken together, we argue that the supplemented analysis of the demagnetization Fourier periodicity, in addition to the analysis of the Fourier amplitude, serves as conclusive evidence for the modulation of demagnetization dynamics by plasmonic excitation.

Based on the discussion above, we have revised certain statements in the original manuscript to emphasize the importance of the correlation between SPP and demagnetization (line 328-337, 352-357 in the revised manuscript highlighted in green), and the observed additional magnetic periodicity (line 362-365, 378-379 in the revised manuscript highlighted in green) that serves as evidence for the claim that the demagnetization dynamics is influenced by the SPP excitation/propagation.

In fact, one can simply “quench” the SPP field by switching the polarization of the pump light to be surface parallel, in which case only the optical grating exists and no SPP should be excited. In this case, one is allowed to compare the TM and TE excitation, and differentiate the plasmonic contribution in space and time. I think this is a more direct evidence of plasmonic effect than the numerical simulation and analysis in the rising time.

Thank you for your insightful suggestions. We agree that comparing the experimental results of the demagnetization dynamics with and without SPP excitation provides more direct evidence on how plasmonic excitation influence demagnetization dynamics, as opposed to solely comparing with the simulated results for quenched SPP fields as in the original manuscript. We therefore conducted a similar experiment as you proposed.

In the original manuscript, we control the excitation of SPs from the edge connecting the sample and a slanted face of the TEM grid that we use as mirror. To make a straightforward comparison of the demagnetization dynamics in a system where the SPP field is ‘quenched’, we designed an experiment without this edge (plasmonic hotspot). The plasmonic hotspot was eliminated by separating the last mirror used for generating the TG from the sample by $\sim 50 \mu\text{m}$. While keeping all other experimental parameters constant, as shown in the sketch below (Fig. R2(b)).

Figure R2: Schematic illustration of the experimental setup for (a) SPP ‘on’ and (b) SPP ‘off’. The distance between the detached mirror and the sample in (b) was measured by scanning electron microscopy (SEM). (c) displays the spatial distribution of the relative maximum demagnetization amplitude ΔF_{max} with ‘SPP on’ (orange data) and ‘SPP quenched’ (purple data). To better illustrate the contribution of SPP to the spatially dependent demagnetization, we subtracted our simulated reference data (simulated distribution of relative ΔF_{max} dominated by transient optical grating (TG)) in the analysis of both demagnetization results. This subtraction eliminates the spatial influence of pump light (TG) on the demagnetization. The vertical arrow indicates the SPP contribution to the demagnetization.

In the absence of an edge (**Figure R2b**), the optical absorption of the sample can be considered spatially uniform, and the demagnetization results are only influenced by the optical grating (no SPP excitation). This experimental geometry is practically the same as TE excitation.

Following the suggestion of the reviewer, we compared the experimental demagnetization dynamics with and without a plasmonic hotspot (‘SPP on’ versus ‘SPP quenched’). The comparison is shown in

Figure R2c. Note: to more clearly illustrate the contribution of SPP to the local demagnetization, both demagnetization results are subtracted with a simulated reference distribution of relative maximum demagnetization amplitude ΔF_{max} from excitation by a TG (with no SPP excitation). Thus, values around zero on the scale in Figure R2 are similar to the simulated reference results for a model not including SPP excitation.

As shown in the blue data in Figure R2c, we find that in the control experiment (without SPP excitation) the spatial distribution of relative ΔF_{MAX} is close to zero at all spatial coordinates (after subtraction of the simulated reference). This shows: 1) that the local demagnetization dynamics are governed solely by the interference of the pump light and 2) it validates the accuracy of the simulations. From the data we can also observe that no 17 μm beat periodicity is present in the control experiments.

These results are in striking contrast with results from the experiments with ‘SPP on’. Here the spatial distribution of relative ΔF_{max} after subtraction of the simulated reference clearly deviates from zero at spatial coordinates close to the edge. Further, an additional magnetic periodicity of 17 μm is observed. Both these findings are clear signatures of SPP mediated demagnetization. To conclude, our experimental results now distinguish the effects rendered by plasmonic excitation on the spatial distribution of demagnetization magnitude and provide strong evidence for the conclusion that SPP mediates demagnetization dynamics.

We appreciate your suggestions that we believe have greatly improved our manuscript.

We have revised the manuscript to reflect these points clearly at lines 233-249, 285-303, and Figure 4(d) and (f) in the revised manuscript with main text and figure caption highlighted in green, as well as in the revised supplementary information highlighted in green.

To improve the reading experience of the article, and to highlight the modulation of demagnetization by SPP, we have **restructured** the manuscript extensively.

In the original manuscript, the characterization of the spatial distribution of SPP using PINEM (originally Fig. 4) was discussed after the spatiotemporal distribution of demagnetization. In the revised manuscript, this section has been moved to precede the discussion of the spatiotemporal distribution of demagnetization (now Fig.3), as shown in the lines 151-209 and figure caption both with the blue text color in the revised manuscript, as well as in the revised supplemented information with the blue text color. This reorganization makes the discussion section more concise, focusing solely on the explanation of the mechanism of SPP-mediated demagnetization. Additionally, the original manuscript's discussion section had a less clear description of the correlation between the spatial distributions of the SPP intensity and the demagnetization magnitude. In the revised version, we have moved this content to the first paragraph in the discussion section, as shown at lines 328-357 with blue text color in the revised version. We have improved this description, as highlighted in green in lines 328-337, 352-357 in the revised version, as well as in the supplementary information highlighted in green.

2. The conclusion of plasmon induced spin currents is also far reaching. One cannot simply rule out the propagation effect and assign the delayed rising to be due to the spin currents. The spin of SPPs is locked with their momentum, so if an opposite spin configuration is excited by counter propagating SPPs, then one can propose the spin current mechanism. I suppose this could be done with a Lorentz TEM. Simply arguing from the intensity and time perspective is rather weak.

We than the reviewer for his insightful comments/suggestions. We understand the concerns regarding our attribution of the observed maximum demagnetization time delay to plasmon-induced spin currents. Indeed, our experimental results do not demonstrate the presence of spin currents; instead, they were inferred based on existing literature and models. We also acknowledge that the influence of SPP

propagation cannot be ruled out. Therefore, we agree that our original submission did not provide a convincing explanation for the experimental observation of delayed time for reaching maximum demagnetization at spatial coordinates far from the plasmonic hotspot (edge). In the following response, we address your comments in detail.

We appreciate your suggestion for an experiment with counter-propagating SPP excitation that could generate a spin current through opposite spin configurations. In our original manuscript, we introduced the spin current model since the time difference caused by SPP propagation (0.16 ps) was insufficient to fully explain the observed 0.4 ps difference in time for reaching maximum demagnetization. This leaves a time gap of 0.2 ps. In this response, we reexamine the SPP-induced spin current model according to analysis in recent papers [*Phys. Rev. B* 101, 161404 (2020), *Phys. Rev. B* 102, 125431 (2020)] to quantitatively verify/reject whether the spin current model chosen in our original manuscript could explain the accelerated demagnetization time. We have calculated the spin current generated by SPP in our experiments and estimated its contribution to the overall demagnetization amplitude. The detailed description of the theories and calculations has been added in a separate document attached to the response letter. In short, we obtained a calculated vertical (out of plane) spin current in our experimental geometry in the order of 10^5 A/m². However, due to the short time scales involved, this magnitude of spin current implies that it only contributes to approximately 10^{-6} % of the total demagnetization amplitude. Therefore, its contribution to the total demagnetization is negligible, indicating that the spin current model fails to significantly accelerate demagnetization.

In reply to comment 1, we supplemented our study with a control experiment where the SPP excitation was quenched. This experiment not only showed the contribution of SPP to the spatial distribution of demagnetization amplitude but also revealed its contribution to the spatial distribution of demagnetization time, as shown in Figure R3(a). By comparing the results with those from the control experiment, it was evident that the spatial delay at maximum demagnetization time only occurs when the demagnetization process is mediated by SPP propagation. This indicates that the spatial delay in maximum demagnetization time is related to the propagation of SPP. Furthermore, it is clear that the relative time delay at maximum demagnetization increases monotonically with the SPP propagation distance, as illustrated by the orange fit line in Figure R3(a). In our original manuscript, we only considered the time required for SPP propagation, which is related to the SPP group velocity. However, we overlooked any SPP pulse broadening caused by dispersion and damping during SPP propagation. As the SPP wave packet propagates, it broadens and attenuates through dispersion and dephasing [*Nano Lett.* 7, 2, 470–475 (2007)]. Yanan Dai et al. [*ACS Nano* 12, 7, 6588–6596 (2018)] observed broadening of a SPP wave packet in the propagation direction when SPPs were excited from an edge of a silver sample. Our experimental geometry resembles that in this reference [*ACS Nano* 12, 7, 6588–6596 (2018)] and our permalloy sample exhibit higher dielectric loss compared to Ag. Therefore, SPP wave packets will in our system inevitably broaden during propagation. We used the atomistic spin dynamics (ASD) model implemented in Vampire software to simulate how broadening of the excitation pulse length will influence the time to reach maximum demagnetization. The results of the simulations are shown in Figure R3(b)-(c):

Figure R3 (a) Spatial distribution of the relative time delay for reaching maximum demagnetization amplitude for experiments 'with SPP' (orange data) and 'without SPP' (purple data). The arrow refers to the SPP contribution. The lines are a fit to the data. (b) Demagnetization curves excited with increasing pulse width, simulated by using atomistic spin dynamics (ASD) model. Note that the centers of all excitation pulses have been aligned at same time delay. (c) The relative delay at maximum demagnetization time as a function of excitation pulse duration.

From the results in Figure R3(c), we can observe that a broadening of approximately 100 fs is sufficient to result in an additional 0.2 ps delay before reaching maximum demagnetization. This accounts for the remaining 0.2 ps (with ~ 0.2 ps from the time required for SPP propagation). We overlooked the impact of pulse broadening during SPP propagation on demagnetization time in our original manuscript.

In conclusion, after conducting calculations we rule out the significance of the SPP-induced spin current model proposed in our original submission. We therefore investigated the influence of pulse broadening during SPP propagation on the time to reach maximum demagnetization. The results of these new simulations allow us to propose SPP pulse broadening and propagation as the main processes behind the observed time delay at to reach maximum demagnetization.

In light of the above discussion, we in the revised the manuscript sections explaining the observed spatial dependence of maximum demagnetization time delay as a combined contribution of the time delay caused by SPP propagation determined by group velocity and the SPP wave packet's broadening during propagation. These descriptions are added at lines 385-405 of the revised manuscript and highlighted in green, as well as in the corresponding supplementary information highlighted in green.

Minor comments:

The authors should treat abbreviations more carefully. LUEM, UEM, LUTEM are used interchangeably. In addition, some concepts are not well defined, and some are repeatedly defined in the main text. This hinders efficient reading of the manuscript.

Thank you for your kind remind. We have carefully reviewed and addressed the issues regarding abbreviations and concept definitions in the revised manuscript. Specifically, we have ensured that all abbreviations are used consistently throughout the text, especially for the LUEM and TMG abbreviations. Additionally, we have thoroughly checked all concepts to ensure they are well-defined and not repeatedly defined. We hope these modifications meet your expectations and improve the overall quality of our work.

Thank you once again for your constructive feedback. We have carefully addressed each of your concerns in our revisions and hope that these updates clarify our intentions and findings. We believe that the modifications made, and the additional experimental results provided not only address your comments but also strengthen the manuscript considerably. We trust that the changes meet your expectations and enhance the overall quality and clarity of our work.

Reviewer #2 (Remarks to the Author):

Y. Fan and his colleagues have conducted an experimental study on the spatiotemporal dynamics of ultrafast demagnetization in permalloy, facilitated by propagating surface plasmon polaritons (SPP). The experimental technique employed was Lorentz ultrafast electron microscopy, where a transient grating was induced by two different pulse trains. The presence of SPP resulted in enhanced and accelerated demagnetization dynamics, attributed to the field enhancement. Interestingly, in addition to the modulation of the transient grating, a spatial modulation was also observed and explained as interference between the electric field of the laser pulses and the SPP field.

Despite the extensive research and published models on ultrafast demagnetization, the main fundamental questions remain unanswered. The authors utilized a sophisticated and challenging experimental approach, which allowed them to uncover a novel aspect of this phenomenon. The observations are robust and well-understood by the authors. I think that the content of this manuscript may be suitable for publication in Nature Communications. However, I cannot recommend the present version of the manuscript. If the author can revise it, I will be glad to see it again to assess its suitability for Nature Communications.

We are grateful to the reviewer for the valuable insights provided on our manuscript. We appreciate the time and effort you have dedicated to reviewing our work.

We appreciate your positive comments regarding the novelty and robustness of our experimental approach and results. Your comments have been crucial in highlighting areas where our manuscript could be improved to meet the high standards of Nature Communications.

In response to your suggestions, we have optimized the manuscript by removing redundancies and moving detailed technical descriptions to the supplementary materials, while also clarifying the perspectives of our study in the field of magnetoplasmonics and ultrafast magnetism. These revisions enhance the manuscript's clarity, conciseness, and its potential impact in the field.

We hope these changes meet your expectations and make the manuscript suitable for publication in Nature Communications.

1. While the results presented are technically sound and well-presented, they lack context within the existing literature. The authors state that their findings encourage further exploration of ultrafast plasmon-assisted magnetization dynamics (last line of the paper), but fail to disclose the potential scenarios and next steps that may arise from this discovery. Without addressing these questions, the authors' results may appear incremental rather than offering novel perspectives in the field of ultrafast magnetism. Therefore, it is imperative that efforts be made to address these gaps and provide a more comprehensive understanding of the implications of this research.

Thank you for your valuable suggestion regarding the placement of our research within the existing literature. We agree and appreciate your point that our results require deeper discussion within the context of existing literature and our manuscript needs a clearer discussion of potential future scenarios and specific next steps stemming from our findings.

First, in response to your comment that "they lack context within the existing literature," we would like to clarify that our work is not isolated. It is closely connected to the fields of magnetoplasmonics and ultrafast magnetism (*described in Introduction paragraph 1 (ultrafast magnetism) and paragraph 2 (magnetoplasmonics)*). Our findings not only reveal for the first time the mechanism of propagating SPP mediated ultrafast demagnetization dynamics, but also provide new perspectives and methods for characterizing the interaction between femtosecond light pulses and spins at nm spatial dimensions. This is elucidated in two key aspects: on one hand, our paper opens possibilities for investigating light-spin interaction at the nm scale by deliberate tailoring of light, such as introducing optical angular momentum to study its spatial interaction with spin structures. From this perspective, our results can

promote the integration of the photonics field with ultrafast magnetism, expanding the scope of research and application by introducing confined light and structured light in the field of light-magnetism interactions. Secondly, our results demonstrate the potential of using Lorentz ultrafast electron microscopy (LUEM) to detect localized photomagnetic interactions, offering a novel characterization method for studying magnetoplasmonics in the photomagnetic domain. To date, observation of ultrafast magnetodynamics at nanoscale dimensions in *real-space* remain challenging (*Nat Commun* 12, 6337 (2021)), with only a limited number of studies reported. From this perspective, our paper also stands at the forefront of magnetodynamics characterization, specifically in the capabilities of LUEM in characterizing spatially resolved ultrafast magnetodynamics at the subwavelength scale. Therefore, our methodology can gain broader adoption and stimulate further research and development in magnetoplasmonics/ultrafast magnetism research.

Specifically, our findings may inspire research in the following directions, **to name a few**:

(1) THz nanomagnonics

Magnons offer unique advantages: e.g. nm wavelengths at GHz–THz frequencies and the absence of Joule losses. This makes magnons a promising alternative for integrated low-power analog and digital computation, and signal processing (*The 2024 magnonics roadmap, J. Phys.: Condens. Matter* 36 363501 (2024)). The excitation of magnons via plasmons open possibilities for energy-efficient THz magnetic resonator devices and potential development of hybrid devices leveraging both light and magnetic waves for advanced information processing and computing technologies. Furthermore, the ability of plasmonics to focus light down to nm dimensions allows for more precise spatial control over magnon dynamics. There are reports that use the effective magnetic fields generated by SPP to excite localized sub-THz spin precession dynamics in dielectric garnets (e.g. *Nano Lett.* 18, 5, 2970-2975 (2018)). However, research on SPP-induced propagating magnons with non-zero wavevector and their interactions is still lacking. Our subsequent experimental results (outside the scope of this report) demonstrate that our setup described in this paper has the capacity to excite and detect propagating spin waves. Therefore, our results may further advance the study of utilizing SPP to excite, control and detect THz spin waves in different magnetic structures that support light-accessible THz magnons, including various dielectric and antiferromagnetic materials, facilitating their investigation and application.

(2) nanoscale control of all optical switching (AOS)

AOS is actively pursued in the field of magnetic storage, mostly due to the tantalizing prospects of outperforming traditional magnetic field-induced magnetization reversal. Using plasmonics to manipulate AOS has the potential to enhance energy conversion efficiency and provide more precise control (higher density) over magnetic bits. For example, plasmonic gold nanoantennas have been used to locally enhance the optical field and achieve repeatable AOS in TbFeCo films at below 53 nm in the spatial dimension (*Nano Lett.* 15, 10, 6862-6868 (2015)). It is anticipated that further confining light (for AOS applications) to even smaller spatial dimensions may achieve energy dissipation as low as 10 fJ per magnetic bit (*Phys. Rev. B* 86, 140404 (2012)). Therefore, nanoscale resolution for detection and control while maintaining femtosecond time excitation is crucial for future developments within AOS research.

Furthermore, using grating techniques is an alternative method to study AOS dynamics at the nanoscale. Successful observation of AOS dynamics with periods as small as 87 nm in GdFe materials has already been achieved (*Nano Lett.* 22, 4452-4458 (2022)). It is expected that further reducing the grating spacing to be comparable to or smaller than the characteristic diffusion length of spin-polarized currents will reveal novel AOS mechanisms, where ultrafast lateral transport processes are expected to ultimately compete with AOS. Moreover, due to the in-plane magnetic sensitivity of Lorentz UEM, combining our method with grating techniques may be extended to studies of magnetic domain wall dynamics induced

by localized AOS within optical gratings. Thus, our results are anticipated to stimulate more research on nanoscale control of AOS, laying the groundwork for future studies of magnetism on the length and time scales of exchange interactions.

Exploring the nanoscale spatial dependence of magnetization dynamics, a degree of freedom that has been scarcely investigated so far (*ACS Photonics* 3, 8, 1385–1400 (2016)), is also a promising approach to obtain complete understanding of the mechanisms involved in light-spin interactions, such as unveiling intricately ultrafast demagnetization mechanism and light induced **magnetic topological structure dynamics**. The high spatial resolution of LUEM magnetization analysis, combined with tailoring/structuring of light demonstrated may pave the way for nanoscale studies of light-magnetism dynamics. Our pioneering work is expected to inspire further exploration of light-spin/magnon interactions at nanoscale levels, as discussed above.

We added lines a abbreviated/condensed version of the above paragraphs to lines 454-458 (highlighted in yellow colour) in the revised manuscript. This with the purpose to provide a perspective of our work and to explain the significance of our research. We believe these additions contribute to offering novel perspectives from our results in the fields of magnetoplasmonics and ultrafast magnetism.

Thank you again for your valuable suggestions.

2. The paper's length and technical language make it difficult to read and do not align with the broad audience of Nature Communications. The main story line is often lost in sections overloaded with technical details, with a summary only provided in the last sentence. For example, lines 284-287 convey the main point, rendering the preceding paragraph unnecessary. Similarly, lines 202-206 provide a concise summary of a large portion of text above them. I recommend that the authors condense the text by retaining only the essential sentences and relegating technicalities to the supplementary materials or methods. In particular, the section on "mapping the spatiotemporal demagnetisation dynamics" should be revised.

Thank you for pointing this out. We have revised the manuscript to address your concerns by condensing the text and simplifying the technical language. Technical details have been moved to the supplementary materials, as highlighted in yellow in the revised manuscript and supplementary information. We have also thoroughly revised the "mapping the spatiotemporal demagnetisation dynamics" section to enhance clarity and focus, as shown in lines 210-324 in the revised manuscript. We believe these changes significantly improve the readability of the manuscript.

3. Similarly, the description of data analysis centered around Equation 1 can be moved from the main text, with only the demagnetisation time mentioned.

Thank you for this comment. We have moved the description of the data analysis associated with Equation 1 to the supplementary materials, as suggested.

4. Some crucial information is either missing or not properly placed. For instance, the "Results" section should mention the photon energy, type of laser system (oscillator or amplified) and fluence used in the experiment. This would provide readers with a general understanding of the light-matter interaction regime: is a weak perturbation sufficient to induce the observed effect, or is a high energy per pulse necessary?

Thank you for your comment. We have revised the "Results" section to include information of the photon energy (1.2 eV) and fluence ($\sim 3 \text{ mJ cm}^{-2}$) used in the demagnetization experiments. This is shown at lines 104-106 in the revised manuscript (highlighted in yellow).

5. Lines 229-230 mention "Fig. 4c" followed by "Fig. 4b," but it appears that only Fig. 4b is being described.

We apologize for the unclear description. We have revised the manuscript to clearly indicate that the description refers to Fig. 4c, as shown in line 180 in the revised manuscript highlighted in yellow.

6. Line 249: The authors mention the gyration vector \mathbf{g}_x , defining it as a magneto-optical parameter proportional to the magnetisation. Could they clarify this point further? Is \mathbf{g}_x a linear magneto-optical tensor, similar to or exactly the one describing the Faraday rotation, or is it something else?

Thank you for your question. The application of a magnetic field makes the dielectric tensor $\hat{\epsilon}$ asymmetric and dependent on the magnetization of the material. Considering an isotropic medium with $\epsilon_{xx} = \epsilon_{yy} = \epsilon_{zz} = \epsilon$, the electric induction \mathbf{D} can be written as

$$\mathbf{D} = \epsilon \mathbf{E} + i \mathbf{E} \times \mathbf{g}$$

Where $\mathbf{g} = (g_x, g_y, g_z)$ is the gyration vector, which is a pseudovector equals to $\mathbf{g} = \epsilon \times Q \times \mathbf{m}$, $Q = i(\epsilon_{ij}/\epsilon_{ii})$ is the magneto-optical constant, $\mathbf{m} = \mathbf{M}/M_{sat}$, where M_{sat} is the magnetization value at the saturation and m_x , m_y and m_z are the director cosines of \mathbf{M} .

In this case the dielectric tensor takes the following form

$$\hat{\epsilon} = \begin{pmatrix} \epsilon & ig_z & ig_x \\ -ig_z & \epsilon & -ig_y \\ -ig_x & ig_y & \epsilon \end{pmatrix}$$

Therefore, the gyration vector \mathbf{g} is a linear magneto-optical tensor as the one describing the Faraday effect or magneto-optic Kerr effect since both effects result from the off-diagonal components of the dielectric tensor $\hat{\epsilon}$. We have clarified this point in the manuscript to ensure a better understanding.

Finally, we can confidently state that **we have addressed every comment from the reviewers**. In the revised manuscript, the content highlighted in blue indicates changes are made to improve the flow and coherence after restructuring and modifying the article. We sincerely thank the two reviewers for their constructive comments. We believe that their suggestions have significantly improved the readability and credibility of our manuscript, as well as enriched the perspectives brought by our results.

Response letter

In this letter we respond to comments made by the reviewer (with comments in blue font). Our response is following each reviewer comment (in black font).

Reviewer #1 (Remarks to the Author):

I appreciate that Fan et al. have done further experiments to support their conclusions, and have significantly revised their manuscripts.

Regarding queching of SPPs, although the experiment was conducted in a different way than I suggested, i.e. by varying the excitation polarization, it still serves the same purpose in practice. So it is indeed interesting to see the effect of the enhancement of the spin dynamics with and without SPP excitations, and particularly the beating effect.

We sincerely appreciate your review of our work and your valuable suggestions regarding both our experiments and analysis. Your feedback has been very helpful in improving the manuscript. We are also pleased to note that you acknowledge the principal/main finding of our paper. (*‘So it is indeed interesting to see the effect of the enhancement of the spin dynamics with and without SPP excitations, and particularly the beating effect.’*)

As to the delayed spin response, the authors changed their argument to broadening effect of the SPP pulse. This argument seems reasonable qualitatively, but I think it is a material specific question (also the authors compared the results with that in Ag), and needs to be considered along with the propagation effect in my first comment.

With this said, I think it is better to calculate the wavelength, the propagation length, as well as the chirp of the SPPs, based on the dielectric function of the magnetic film. This calculated propagation length will further strengthen the argument for Fig. 3c, and the chirp of SPPs will support whether 100 fs broadening is the correct value to consider.

Lastly, if SPP broadening was considered, perhaps the broadening of the excitation needs to be considered as well.

We greatly appreciate your further review of our work and the valuable suggestions. To summarize, we will in this answer show simulations of the SPP wavelength, SPP propagation length, SPP chirp, and the contribution from the experimental excitation geometry to the delayed spin response we observed. We will address each of these points in our response below. We willingly admit that the delay in maximum demagnetization observed with increasing has been difficult to explain (especially since delays are so short). But now we believe we have finally arrived at a solid explanation, after revisiting the experimental geometry and our data analysis methods.

First, we calculated the SPP wavelength and propagation length. In Fig. S4 of the Supplementary Material, we discussed the calculation of the SPP dispersion relationship for the material system used in the

experiment, namely the vacuum/Al (5nm)/Ni₈₀Fe₂₀ (50nm)/Si₃N₄ (50nm)/vacuum structure. These calculations were performed using the Radio Frequency Module in COMSOL Multiphysics, based on the Finite Element Method (FEM). We determined that the SPP wavevector, k_y , stimulated at the top interface (Air/ Al (5 nm)/ Ni₈₀Fe₂₀ (50 nm)) at 1.2 eV is $6.38 \times 10^6 \text{ m}^{-1}$ (The contribution of SPP from the bottom interface can be neglected due to the strong light absorption in the sample (Fig. S4e)). Following this, we used the same method, material system, and laser parameters to calculate the SPP propagation length. A comparison between the calculated and experimental results is shown in Fig. R1 below. Fig. R1(a) shows a schematic of the simulated SPP intensity attenuation during propagation by using COMSOL. Fig. R1(b) presents the simulated spatial distribution of the relative SPP intensity over a distance of 50 μm , while Fig. R1(c) displays the PINEM-measured relative SPP intensity over the same distance, as depicted in Fig. 3c of the original manuscript. We fitted the two curves in Figs. R1(b) and R1(c) using an exponential decay function, $y = A_I \times \exp(-x/t_I) + y_0$, where the fitted parameter t_I represents the SPP propagation length (i.e., the distance at which the intensity decreases to 1/e of its initial value). Through this fitting, we find that the SPP propagation lengths obtained from Fig. R1(b) and Fig. R1(c) are comparable.

Fig. R1 (a) Schematic illustration of simulated SPP attenuation during propagation under relevant experimental conditions. For the sake of clarity we only show the first 25 μm segment here. The inset provides an enlarged view of the simulated SPP propagation in the selected region of (a). (b) and (c) shows the spatial distribution of the simulated and experimental relative SPP intensity, respectively. In (c), the data corresponds to the PINEM data from Fig. 3(c) of the main manuscript. The fitted propagation lengths from the simulation and the PINEM measurements are in agreement.

Next, we calculated the SPP chirp using the expression: chirp = propagation distance \times GVD (1.2 eV) \times (bandwidth) (where the bandwidth, compared to the curvature of the SPP dispersion curve, is relatively small). Here, $\text{GVD}(\omega_0)$ represents the group-velocity dispersion $\frac{\partial^2 k}{\partial \omega^2}$ at 1.2 eV. Substituting the SPP propagation distance of 50 μm and the GVD value of $1.1073 \times 10^{-25} \text{ s}^2/\text{m}$ (obtained from the SPP dispersion relationship in Fig. S4(d) of the Supplementary Material), along with the experimental optical wavelength of $1024 \text{ nm} \pm 7 \text{ nm}$, corresponding to a frequency bandwidth of $\Delta\omega = 2.64 \times 10^{15} \text{ rad/s}$, we calculated the SPP chirp to be $1.46 \times 10^{-18} \text{ s}$. This result shows that the SPP chirp cannot explain the 100 fs pulse broadening. We are grateful to the reviewer for helping us rule out the possibility of the SPP chirp being responsible for the delayed spin response.

To investigate the source of the 100 fs broadening, we followed the reviewer's suggestion and calculated the excitation laser broadening resulting from the optical path difference from the two laser paths. As shown in Fig. S2 of the Supplementary Material, the optical path difference between the reflected beam and the direct beam causes pulse broadening in the resulting interference pattern at the sample surface. We calculated the FWHM (full width at half maximum) of the exciting beam resulting from the interference as a function of the optical path difference of the two beams (illustrated in Fig. R2(a)). We found that at spatial locations 50 μm from the edge (at an optical path difference of approximately 50 phases), the FWHM of the excitation pulse increases by ~ 40 fs. According to the calculations from our previous response letter regarding the effect of pulse broadening on the demagnetization curve, as shown in Figs. R2(b)-(c), this excitation laser broadening results in a maximum demagnetization time delay of around 0.1 ps. In addition, and as previously discussed, the pulse center of the exciting beam (from interference) is also delayed due to the optical path difference, which affects the onset time of demagnetization. Our calculations show that the pulse center of the interfered beam is delayed by approximately 0.1 ps at a distance of 50 μm from the edge, leading to an additional 0.1 ps delay in the maximum demagnetization time. Therefore, the excitation laser pulse broadening and pulse center delay due to optical path difference contribute to a total maximum demagnetization time delay of approximately 0.2 ps over the distance of 50 μm from the edge. This is a correction of our initial interpretation that the contribution from the broadening of the excitation pulse to the maximum demagnetization time delay was negligible, as recalled in Fig. R2(d)-(f).

After re-examining our previous response, we realized that the absence of a noticeable delay in the maximum demagnetization time caused by the excitation laser was due to the relatively large offset (7 μm) used in the FFT slicing method, as shown in Fig. R2(e). This resulted in fewer data points for fitting, which failed to accurately reflect the trend in the data. The spatial distribution of the maximum demagnetization time in the control experiment (without plasmonic hotspots) is shown by the purple data in Fig. R2(f). In our earlier response, only 7 demagnetization curves were fitted within a 50 μm range. To improve the accuracy of the data fitting, we reanalyzed the data using a smaller FFT slice offset (1 μm), as illustrated in Fig. R2(g). With this adjustment, we were able to fit 50 demagnetization curves within the same 50 μm range. The resulting spatial distribution of the maximum demagnetization time is represented by the purple data in Fig. R2(h). For comparison, the orange data in Fig. R2(h) shows the same spatial distribution of the maximum demagnetization time with a plasmonic hotspot, as the orange data in R2(f). After applying this refined FFT slice offset, we observed that the control experiment (without plasmonic hotspots) also exhibited a noticeable spatial delay in the maximum demagnetization time, occurring ~ 0.2 ps earlier at the edge compared to 50 μm away, consistent with our calculations.

Additionally, we noted that after applying this more refined fitting, the slope difference between the two sets of comparison data in Fig. R2(h) became nearly indistinguishable (the slope of the purple data is $(5.58 \pm 1.46) \times 10^{-3}$ ps/ μm , while the slope of the orange data is $(7.92 \pm 2.12) \times 10^{-3}$ ps/ μm). To further unify the offset values used between the two data sets shown in Fig. R2(h), we reanalyzed the orange data in Fig. R2(h) to equalize the FFT slice offsets of both data sets. The results are presented in Fig. R2(i). We observed that, after applying the same FFT slice offset (1 μm) to both data sets, the slope of the purple data (without plasmonic hotspot) is $(5.58 \pm 1.46) \times 10^{-3}$ ps/ μm , while the slope of the orange data (with plasmonic hotspot) is $(5.83 \pm 0.74) \times 10^{-3}$ ps/ μm . The negligible difference between the two slopes suggests that the primary source of the spatial delay in the maximum demagnetization time observed with plasmonic hotspots, as reported in the original manuscript, is the same as in the control experiment without plasmonic hotspots—namely, the optical path difference of the transient optical grating.

In our last response letter, we also mentioned that under our experimental conditions, SPP propagation over 50 μm , determined by the SPP group velocity, would take approximately 0.2 ps. However, because the SPP and the excitation laser propagate inward from the edge simultaneously, the total delay in the maximum demagnetization time cannot be simply summed. Here, we believe that, due to the time resolution constraints of our instruments, we are unable to separate the influence of the pure SPP propagation effect on the delay in the maximum demagnetization time from the effects of the excitation laser. Therefore, we conclude that the spin response delay observed in the original manuscript is primarily caused by the optical path difference of the excitation laser.

Fig. R2 (a) Influence of the interfered pulse FWHM with laser propagation length; (b) Demagnetization curves excited with increasing pulse FWHM, simulated by using atomistic spin dynamics (ASD) model. Note that the pulse centers (in time) of all excitation pulses have been aligned at same time delay; (c) The relative delay at maximum demagnetization time as a function of excitation pulse FWHM. (d) Schematic illustration of the experimental setup without a plasmonic hotspot. (e) Diagram of the FFT slicing method used in our first response letter, where the slice offset we used was 7 μm . (f) Spatial distribution of the maximum demagnetization time from the previous analysis for experiments ‘with SPP’ (orange data) and ‘without SPP’ (purple data) in our first response letter. The orange data has been offset along the y axis for clarity. (g) Reanalysis of the ‘without SPP’ data using a more refined FFT slicing method, with a slice offset of only 1 μm . (h) Spatial distribution of the maximum demagnetization time obtained using the more refined FFT slicing method for experiments ‘without SPP’ (purple data), while the data for experiments ‘with SPP’ (orange data) remain unchanged. It can be seen that, with more data points used for fitting, the experimental data ‘without SPP’ (purple data) now also show a weak maximum demagnetization time delay. The orange data has been offset along the y axis for clarity. (i) The orange data ‘with SPP’ is reanalyzed using the same slice offset as the purple data ‘without plasmonic hotspot’ to enable an accurate comparison between the two datasets. The fitted slopes of the two data sets show no observable difference. This indicates that the observable maximum demagnetization time delay in the ‘with plasmonic hotspot’ (orange data) experiment is primarily due to the optical path difference of the two excitation beams. The modulation of demagnetization time caused by SPP propagation is masked by the modulation from the excitation laser, making it undetectable in the present experimental geometry.

We are grateful for the reviewer's comments, which helped isolate an explanation to the observed delay in maximum demagnetization. We would like to stress that in our view this was a peripheral observation in the paper, which main observation is the observed magnetic beat.

Based on the calculations and discussions above, we have made revisions marked in blue text to revised manuscript. The discussion on SPP propagation length has been added to the revised manuscript at lines 274-275 and also in the Supplementary Material in Fig. S4f. The calculation of SPP chirp was not included in the manuscript, as the calculation indicates negligible broadening due to SPP chirp. Additionally, the discussion on excitation laser broadening and pulse center delay due to the optical path difference, as well as its impact on the spatial distribution of maximum demagnetization time, has been added at lines 242-264 of the revised manuscript and in Fig. S10 of the Supplementary Material. The comparison between the spatial distribution of the maximum demagnetization time for the 'without plasmonic hotspot' and 'with plasmonic hotspot' data has been updated in Fig. S9 in the Supplementary Material.

If the authors could supplement these results to further the current manuscript, I think it will be appropriate for publication.

We would like to once again express our gratitude to the reviewer for the professional comments and suggestions. We have completed all the calculations you recommended and have incorporated the relevant results into the manuscript. Your insightful feedback has significantly enhanced the quality and clarity of the paper, and we greatly appreciate the improvements your suggestions have brought to our work.

Reviewer #1 (Remarks to the Author):

The authors have supplemented their experimental observations with further calculations. While the conclusion, which is kind of expected, is somewhat different than before, their experimental observation on the spin beating is still interesting to the community. With the interpretation being corrected and updated, I therefore think it is appropriate for publication.

Answer:

Thank you for your positive evaluation of our manuscript. We sincerely appreciate your constructive feedback throughout the review process, which has significantly enhanced the quality of our work. We are pleased that you acknowledge the additional calculations and updated interpretations we have provided, and that you find our experimental observations on spin beating to be of interest to the community.